# Inhibition of Adipogenesis by Diphlorethohydroxycarmalol (DPHC) through AMPK Activation in Adipocytes

**DOI:** 10.3390/md17010044

**Published:** 2019-01-10

**Authors:** Min-Cheol Kang, Yuling Ding, Hyun-Soo Kim, You-Jin Jeon, Seung-Hong Lee

**Affiliations:** 1Department of Marine Life Sciences, Jeju National University, Jeju 63243, Korea; networksun@naver.com (M.-C.K.); gustn783@naver.com (H.-S.K.); youjinj@jejunu.ac.kr (Y.-J.J.); 2Korea Food Research Institute, 245 Nongsaengmyeong-Ro Iseo-Myeon, Wanju-Gun, Jeollabuk-Do 55365, Korea; 3Department of Pharmaceutical Engineering, Soonchunhyang University, Asan 31538, Korea; dingyuling@naver.com

**Keywords:** adipogenesis, antiobesity, adipocytes, diphlorethohydroxycarmalol (DPHC)

## Abstract

The purpose of this study was to investigate the antiobesity effect and the mechanism of action of diphlorethohydroxycarmalol (DPHC) isolated from *Ishige okamurae* in 3T3-L1 cells. The antiobesity effects were examined by evaluating intracellular fat accumulation in Oil Red O-stained adipocytes. Based on the results, DPHC dose-dependently inhibited the lipid accumulation in 3T3-L1 adipocytes. DPHC significantly inhibited adipocyte-specific proteins such as SREBP-1c, PPARγ, C/EBP α, and adiponectin, as well as adipogenic enzymes, including perilipin, FAS, FABP4, and leptin in adipocytes. These results indicated that DPHC primarily acts by regulating adipogenic-specific proteins through inhibiting fat accumulation and fatty acid synthesis in adipocytes. DPHC treatment significantly increased both AMPK and ACC phosphorylation in adipocytes. These results indicate that DPHC inhibits the fat accumulation by activating AMPK and ACC in 3T3-L1 cells. Taken together, these results suggest that DPHC can be used as a potential therapeutic agent against obesity.

## 1. Introduction

Chronic obesity is one of the most detrimental health issues and a major social problem in the 21st century. In 2017, nearly two billion individuals were reported to be overweight, of which 671 million were obese [1,2]. Excessive food intake and lack of exercise are the most common causes of obesity. Obesity is a metabolic disease characterized by an excessive accumulation of body fat and further associated with complications such as type 2 diabetes, hypertension, hyperlipidemia, and increased risk of cancer and cardiovascular disease [3,4]. Several studies have reported the relationship between obesity and lipid accumulation by evaluating the differentiation of adipocytes. Synthetic antiobesity drugs such as orlistat (Xenical) and sibutramine (Reductil) have widely been used for the treatment of obesity. However, they are associated with side effects including thirst, insomnia, constipation, tension headaches, and steatorrhea [5]. Therapeutic application of natural substances is safer and less toxic than the use of synthetic drugs. Thus, many of the recent investigations have focused on the development of antiobesity agents from natural substances [6,7]. Adipocyte-specific proteins play a crucial role in adipocyte differentiation and lipid accumulation. These include enhancer binding proteins (C/EBP), sterol regulatory element-binding protein 1c (SREBP-1c), peroxisome proliferator activated receptor-γ (PPARγ), adiponectin, perilipin, fatty acid synthase (FAS), fatty acid binding protein (FABP4), and leptin [8]. Adenosine monophosphate-activated protein kinase (AMPK) is a heterotrimeric enzyme, an important mediator involved in regulating energy balance in the human body [9,10]. Many reports show that natural substances could inhibit fat accumulation by suppressing the expression of adipogenic-specific proteins during adipocyte differentiation. Some examples include dioxinodehydroeckol isolated from *Ecklonia cava*, indole derivatives isolated from *Sargassum thunbergii*, ethanol extracts of *Aster yomena*, and *Pinus koraiensis* leaves [11,12,13,14]. 

*Ishige okamurae* is a brown seaweed widespread in South Korea throughout Japan to China. The ethanolic extract of *I. okamurae* has shown antioxidant, antidiabetes, antihypertension, and antiobesity effects [15,16,17]. Recent evidence reported that diphlorethohydroxycarmalol (DPHC) isolated from *I. okamurae* could efficiently induce apoptosis via downregulating Bcl-2 while activating Bax caspase-3 and caspase-9 in adipocyte cells [18]. Ihn et al. reported that DPHC suppresses osteoclast differentiation by downregulating the NF-kB signaling pathway [19]. However, none have studied the inhibitory effects of DPHC upon fat accumulation in 3T3-L1 cells and its molecular mechanism. The present study aimed at investigating the effects of DPHC on adipogenesis in adipocytes. 

## 2. Results 

### 2.1. DPHC Inhibited 3T3-L1 Adipocyte Differentiation and Triglyceride Composition

The MTT assay is widely used in evaluating cell viability and toxicity. The effect of DPHC on 3T3-L1 cell viability was measured by the MTT assay in this study. As shown in Figure 1, these data indicate that DPHC does not affect the viability of 3T3-L1 adipocytes at 12.5, 25, 50, and 100 μM concentrations. The Oil Red O staining assay was used to measure the adipocyte differentiation and lipid accumulation in 3T3-L1 adipocytes. These results indicated that DPHC inhibited the lipid accumulation in 3T3-L1 adipocytes in a dose-dependent manner. These results suggest that DPHC possesses potent antiadipogenic effects due to the inhibition of adipocyte differentiation and adipogenesis.

### 2.2. Effects of DPHC on the Expression of Adipogenic-Specific Protein Levels during the Differentiation of 3T3-L1 Cells

The expression levels of key adipogenic-specific proteins were investigated to elucidate the molecular mechanisms underlying the inhibitory effect of DPHC on 3T3-L1 adipocyte differentiation. The investigated proteins include CCAAT/enhancer-binding protein-α (C/EBPα), sterol regulatory element binding protein-1c (SREBP-1c), peroxisome proliferator-activated receptor-γ (PPARγ), and adiponectin. As shown in Figure 2, DPHC treatment significantly decreased the levels of the adipogenic-specific proteins C/EBPα, SREBP-1c, PPARγ, and adiponectin in adipocytes. Moreover, it is well known that adipogenic-specific proteins could synergistically activate the downstream promoters of adipocyte-specific proteins, including perilipin, fatty acid synthase (FAS), fatty acid binding protein (FABP4), and leptin, that play a critical role in modulating fatty acid synthesis. We measured the inhibitory effect of DPHC on fatty acid synthesis-related proteins in adipocyte cells. These results show that DPHC could significantly inhibit fatty acid synthesis by downregulating adipogenic-specific proteins, including perilipin, FAS, FABP4, and leptin. Resistin, an adipocyte-derived cytokine, may contribute to the development of obesity. This study showed that DPHC treatment decreased the expression of resistin in the adipocytes. Collectively, DPHC acts by regulating adipogenic-specific proteins through the inhibition of fat accumulation and fatty acid synthesis in adipocyte cells.

### 2.3. Effects of DPHC on the Activation of AMPK and ACC in 3T3-L1 Adipocytes

The 5’ adenosine monophosphate-activated protein kinase (AMPK) is a major regulator of whole-body energy homeostasis, which gets activated by lower intracellular ATP levels. Recent reports suggest that AMPK plays an important role in the metabolism of energy, glucose, and ATP production [20]. We investigated the effect of DPHC on the phosphorylation of AMPK and acetyl-CoA carboxylase (ACC) in adipocytes. As shown in Figure 3, DPHC treatment significantly increased both AMPK and ACC phosphorylation in 3T3-L1 adipocytes. These results indicate that DPHC could inhibit fat accumulation by activating AMPK and ACC in 3T3-L1 cells.

## 3. Discussion

Fat accumulation in adipose tissues is a complex process involving a number of different metabolic and signaling pathways. Adipocytes play a vital role in the regulation of energy intake, energy expenditure, and lipid and carbohydrate metabolism [21]. Excessive fat accumulation in adipocytes increases various risk factors including inflammation, hypertension, and heart diseases. Many recent studies have focused on developing antiobesity agents from natural substances, which could inhibit fat accumulation in adipocytes [22,23]. However, few reports are available on the antiobesity effects of polyphenol compounds from seaweeds. In the present study, investigations were done to evaluate the antiobesity effects of DPHC and its mode of action in adipocytes. Fat accumulation is regarded as a regulatory process, whereas a number of adipocyte-specific proteins are involved in the mediation of lipid synthesis, lipolysis, and glucose uptake in adipocytes. These adipocyte-specific proteins get induced during the differentiation of preadipocytes into adipocytes and play an essential role during adipogenesis. The differentiation of 3T3-L1 preadipocytes into adipocytes is mainly controlled by the family of adipogenic-specific factors, including C/EBPα, SREBP-1c, PPARγ, perilipin, FAS, FABP4, and leptin [24,25]. Thus, reducing the expression of adipogenic-specific factors may be an effective strategy for inhibiting fat accumulation in adipocytes. Several recent studies have focused on the potential of polyphenols such as dieckol, epigallocatechin-3-gallate, and resveratrol upon the inhibition of fat accumulation in differentiating 3T3-L1 cells via measuring the decreased expression levels of adipogenic-specific factors [26,27,28,29,30]. Our results indicate that DPHC treatment plays a critical role in inhibiting fat accumulation via decreasing the expression levels of adipogenesis-associated proteins in 3T3-L1 cells. Activation of AMPK increases glucose transport and fatty acid oxidation in adipocytes. Promising strategies for treating obesity include the reduction of fat accumulation through inhibiting adipogenesis-specific proteins and activating the AMPK pathway. A number of researchers have reported that AMPK is a member of the metabolite-sensing kinase family of proteins and plays a central role in regulating glucose and lipid metabolism. It has previously been reported that activated AMPK inhibits lipogenesis and adipocyte differentiation. Furthermore, ACC is a multi-subunit enzyme that regulates enzymes involved in malonyl-CoA production, fatty acid synthesis, and fatty acid oxidation in adipocytes [31,32].

We found that DPHC treatment could significantly increase both AMPK and ACC phosphorylation in adipocyte cells. These results indicated that DPHC inhibits the fat accumulation by activating AMPK and ACC in adipocytes.

## 4. Materials and Methods

### 4.1. Materials

The brown alga *I*. *okamurae* was collected from the coast of Jeju Island, Korea. All collected samples were washed with tap water to remove salt, sand, and epiphytes attached to the surface, followed by careful rinsing with fresh water and were then maintained in a refrigerator at −20 °C. Next, the frozen samples were lyophilized and homogenized with a grinder before the extraction.

Dulbecco’s modified Eagle’s medium (DMEM), fetal bovine serum (FBS), bovine serum (BS), phosphate-buffered saline (pH 7.4; PBS), and penicillin–streptomycin (PS) were purchased from Gibco BRL (Grand Island, NY, USA). All chemicals and reagents used were of analytical grade and obtained from commercial sources. 3-Isobutyl-1-methylxanthine (IBMX), dexamethasone, insulin, and 3-(4,5-dimethylthiazol-2-yl)-2,5-diphenyl tetrazolium bromide (MTT) were purchased from Sigma Chemical Co. (St. Louis, MO, USA). Antibodies to CCAAT/enhancer-binding protein-α (C/EBPα; #2295; Cell Signaling), fatty acid binding protein (FABP4; F2120; Cell Signaling), and adenosine monophosphate-activated protein kinase (AMPK; #2535; Cell Signaling) were purchased from Cell Signaling Technology (Bedford, MA, USA). Antibodies to sterol regulatory element binding protein-1c (SREBP-1c; sc-13551; Santa Cruz), peroxisome proliferator-activated receptor-γ (PPARγ; sc-7273; Santa Cruz), adiponectin, perilipin, fatty acid synthase (FAS; sc-715; Santa Cruz), and leptin were obtained from Santa Cruz Biotechnology (Santa Cruz, CA, USA).

### 4.2. Extraction and Isolation

Dried *I. okamurae* powder was extracted three times with 80% methanol and filtered. The filtrate was rotary-evaporated at 40 °C to obtain the methanol extract, which was suspended in distilled water and partitioned using chloroform. The chloroform fraction was fractionated by silica column chromatography with stepwise elution by a chloroform–methanol mixture (30:1→1:1) to separate the active fractions in the chloroform extract. The active fraction was subjected to further purification using a Sephadex LH-20 column with 100% methanol. The selected active fraction was further purified by reversed-phase high-performance liquid chromatography (HPLC) using a Waters HPLC system (Alliance 2690; Waters Corp., Milford, MA, USA) equipped with a Waters 996 photodiode array detector and C18 column (J’sphere ODS-H80, 250 × 4.6 mm, 4 μm; YMC Co., Kyoto, Japan) by stepwise elution with a methanol–water gradient (UV absorbance detection wavelength, 296 nm; flow rate, 1 mL/min). The eluate was finally purified by high-performance liquid chromatography (HPLC), and the structure of DPHC was determined (Figure 4). The DPHC contents of the 80% methanol extract from *I. okamurae* ranged from ~2 to 3%. The compound was dissolved in dimethyl sulfoxide (DMSO) and employed in experiments in which the final concentration of DMSO in the culture medium was adjusted to <0.01%.

### 4.3. Cell Culture

3T3-L1 preadipocytes obtained from the American Type Culture Collection (Rockville, MD, USA) were cultured in DMEM containing 1% PS and 10% bovine calf serum (Gibco BRL) at 37 °C under a 5% CO_2_ atmosphere. To induce differentiation, 2-day post-confluent preadipocytes (designated as day 0) culture media was replaced with MDI differentiation medium (DMEM containing 1% PS, 10% FBS, 0.5 mM IBMX, 0.25 μM dexamethasone, and 5 μg/mL insulin) for 2 days. The cells were then maintained for another 2 days in DMEM containing 1% PS, 10% FBS, and 5 μg/mL insulin. Thereafter, the cells were maintained in post-differentiation medium (DMEM containing 1% PS and 10% FBS), with the replacement of the medium every 2 days. To examine the effects of test samples on the differentiation of preadipocytes to adipocytes, the cells were cultured with MDI medium in the presence of test samples. Differentiation was measured by the expression of adipogenic markers and the appearance of lipid droplets and was completed on day 8.

### 4.4. Cell Viability Assay

Cytotoxicity of DPHC against 3T3-L1 cells was assessed via a colorimetric MTT assay. 3T3-L1 preadipocytes plated on 24-well plates were treated with DPHC at 37 °C for 48 h. MTT stock solution (100 μL; 2 mg/mL in PBS) was then added to each well to a total reaction volume of 600 μL. After 4 h of incubation, the plates were centrifuged (800× *g*, 5 min) and the supernatant was aspirated. The formazan crystals in each well were dissolved in 300 μL of DMSO, and the absorbance was measured with an ELISA plate reader at 540 nm.

### 4.5. Determination of Lipid Accumulation by Oil Red O Staining

To induce adipogenesis, 3T3-L1 preadipocytes were seeded in 6-well plates and maintained for two days after reaching confluence. The media was then exchanged with differentiation medium (DMEM containing 10% FBS, 0.5 mM IBMX, 0.25 μM Dex, and 10 μg/mL insulin) and cells were treated with test samples. After two days, the differentiation medium was replaced with adipocyte growth medium (DMEM supplemented with 10% FBS and 5 μg/mL insulin), which was refreshed every two days. After adipocyte differentiation, the cells were stained with Oil Red O for measure lipid content. Briefly, cells were washed with PBS, fixed with 10% buffered formalin and stained with Oil Red O solution (0.5 g in 100 ml isopropanol) for 60 min. After removing the staining solution, the dye retained in the cells was eluted into isopropanol and the optical density was measured at 520 nm. Images were collected on an EVOS microscope (ThermoFisher Scientific, Waltham, MA, USA).

### 4.6. Western Blot Analysis

Cells were lysed in lysis buffer (20 mM Tris, 5 mM EDTA, 10 mM Na_4_P_2_O_7_, 100 mM NaF, 2 mM Na_3_VO_4_, 1% NP-40, 10 mg/mL aprotinin, 10 mg/mL leupeptin, and 1 mM PMSF) for 1 h and then centrifuged at 12,000 rpm for 15 min at 0 °C. The protein concentrations were determined by using a BCA^TM^ protein assay kit. The lysate containing 40 μg of protein was subjected to electrophoresis on sodium dodecyl sulfate (SDS)–polyacrylamide gels, and the gels were transferred onto nitrocellulose membranes. The membranes were blocked in 5% nonfat dry milk in TBST (25 mM Tris–HCl, 137 mM NaCl, 2.65 mM KCl, 0.05% Tween 20; pH 7.4) for 1 h. The primary antibodies were used at a 1:1000 dilution. Membranes were incubated with the primary antibodies at 4 °C overnight. Then, the membranes were washed with TBST and incubated with the secondary antibodies (at 1:3000 dilutions). Signals were developed using an Enhanced chemiluminescence (ECL) Western blotting detection kit and exposed to X-ray films.

### 4.7. Statistical Analysis

Data were analyzed using the Statistical Package for the Social Sciences (SPSS) for Windows (Version 8). Values were expressed as means ± standard error (SE). A *p*-value of less than 0.05 was considered significant.

## 5. Conclusions

In conclusion, our data demonstrated that DPHC suppressed adipocyte differentiation and fat accumulation by inhibiting adipogenesis-specific proteins in adipocytes. Taken together, these results suggest that DPHC can be a useful candidate and a potential therapeutic agent for treating obesity.

## Figures and Tables

**Figure 1 marinedrugs-17-00044-f001:**
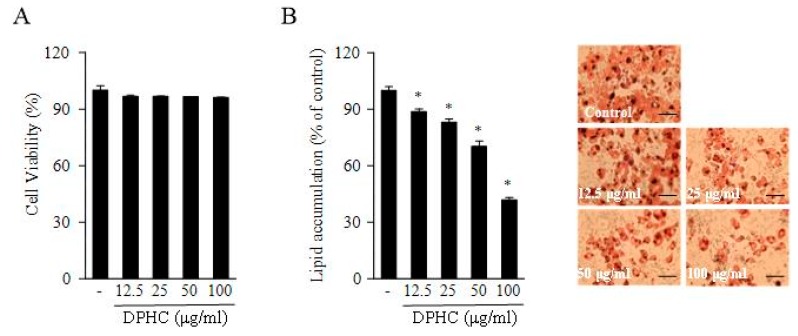
The effect of diphlorethohydroxycarmalol (DPHC) on the cell viability of 3T3-L1 preadipocytes treated for 48 h (**A**). DPHC inhibits intracellular lipid accumulation in 3T3-L1 adipocytes. Lipid accumulation was determined by Oil Red O staining (**B**). Scale bars for B is 50 μm. Data are expressed as the mean of three independent experiments, and the error bars represent the mean ± SE. Significant differences from the control group were identified at * *p* < 0.05.

**Figure 2 marinedrugs-17-00044-f002:**
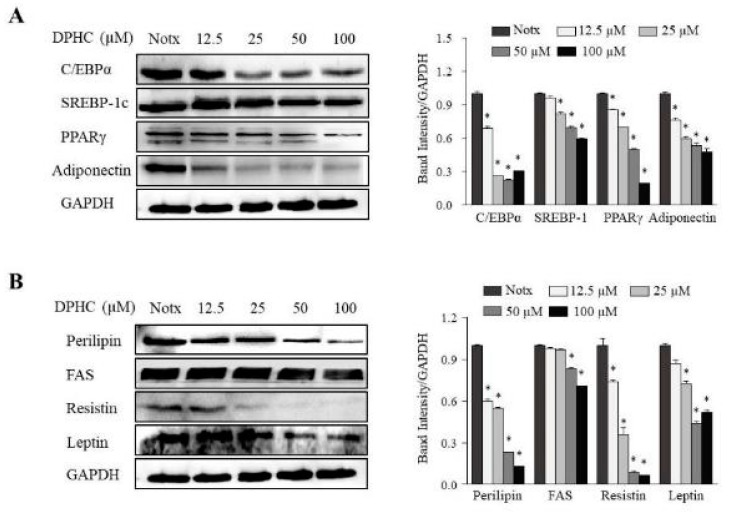
The effect of DPHC treatment on adipogenic-specific protein levels in 3T3-L1 adipocytes. The 3T3-L1 preadipocytes were incubated in a differentiation medium with or without the indicated concentrations of DPHC for eight days (from day 0 to day 8). The expression of sterol regulatory element-binding protein 1c (SREBP-1c), peroxisome proliferator-activated receptor-γ (PPARγ), CCAAT/enhancer-binding protein α (C/EBPα), and adiponectin were assessed by Western blotting (**A**). The expression of perilipin, fatty acid synthase (FAS), fatty acid binding protein (FABP4), and leptin were assessed by Western blotting (**B**). Immunoblot figures are representative of three independent experiments, and each value is expressed as the mean ± SE of three determinations. Significant differences from the control group were identified at * *p* < 0.05.

**Figure 3 marinedrugs-17-00044-f003:**
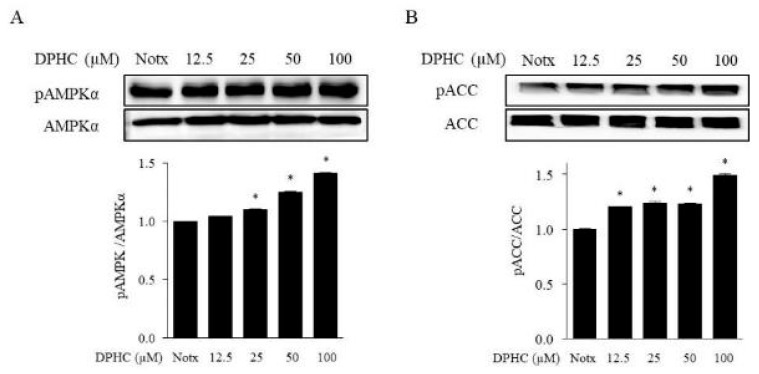
DPHC suppresses the activation of p-AMPKα and p-ACC in 3T3-L1 preadipocytes. 3T3-L1 preadipocytes were maintained in Dulbecco’s Modified Eagle Medium (DMEM) with or without different concentrations of DPHC for eight days (from day 0 to day 8) until their differentiation into adipocytes. Immunoblot figures are representative of three independent experiments, and each value is expressed as the mean ± SE of three determinations. Significant differences from the control group were identified at * *p* < 0.05.

**Figure 4 marinedrugs-17-00044-f004:**
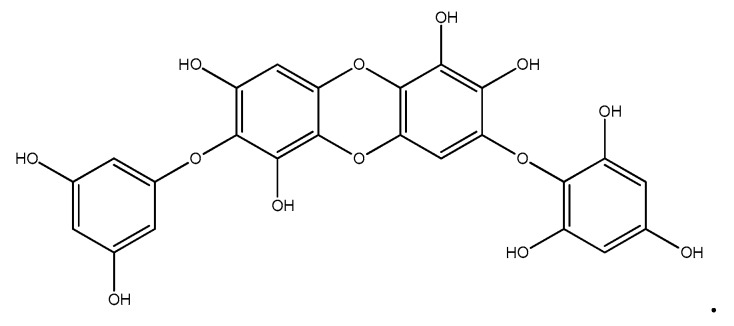
Chemical structure of diphlorethohydroxycarmalol (DPHC).

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
