# Peer review of "Inhibition of Adipogenesis by Diphlorethohydroxycarmalol (DPHC) through AMPK Activation in Adipocytes"

_marinedrugs, 2019, doi:10.3390/md17010044_

Reviewer 1 Report

The manuscript, marinedrugs-410261,describes that   diphlorethohydroxycarmalol (DPHC) isolated from  Ishige okamurae inhibit dipogenesis through AMPK activation in adipocytes. It is interesting thatinhibition of adipogenesis by DPHC with reasonable experiments for support of the author’s discussion.  In this time, this manuscript is  suitable to publish Marine Drugs after the minor revision. 

The  reviewer consider that it is not enough to explain property of DPHC  so that references is required  adding one referenced as below: 

1) International Journal of Molecular Sciences (2017), 18(12), 2635/1-2635/10.

2) The author should add the isolated yield of DPHC in  " 4.2. Extraction and Isolation”. 

 Author Response

Reviewer #1: The manuscript, marinedrugs- 410261,describes that   diphlorethohydroxycarmalol (DPHC) isolated from  Ishige okamurae inhibit dipogenesis through AMPK activation in adipocytes. It is interesting that inhibition of adipogenesis by DPHC with reasonable experiments for support of the author’s discussion.  In this time, this manuscript is  suitable to publish Marine Drugs after the minor revision.
Response: Thank you very much for spending your valuable time in assessing our manuscript. We appreciate you detailed review and salient comments. We have carried out necessary modifications to the manuscript based on your comments.

Comment: The reviewer consider that it is not enough to explain property of DPHC so that references is required adding one referenced as below:
1) International Journal of Molecular Sciences (2017), 18(12), 2635/1-2635/10.
Response: We have added that reference in introduction in page 2 line 56-57.
2) The author should add the isolated yield of DPHC in ” 4.2. Extraction and Isolation
Response: We have added that reference in introduction in page 5 line 169-170
Thank you in advance for your cooperation. I look forward to receiving your kind response.
Sincerely
Seung-Hong Lee

Reviewer 2 Report

I really appreciate the huge effort of the authors for the composition of this manuscript. It deals with the high potential of the role of DPHC on adipogenesis. However, to strengthen these findings, there are a few questions/suggestions that have arisen throughout the reading.

 -The authors describe in Figure 1 the number of replicates and the data representation of the graphs. It would be proper to do the same with other Figures. How do you calculate SE? Do you mean SEM or SD?

-In Figure 2, you have analysed Resistin, but you don’t comment this result anywhere.

-In Figure 3, is it said that DPHC suppresses the activation of p-AMPKa and p-ACC, when it should be “activates”. Moreover, you should clarify that pAMPK phoshorylates ACC inactivating it, and thus blocking downstream effects. This statement is not clear neither in the results nor in discussion.

-Readers should have references for all antibodies used, and phosphorylations analysed, since depending on the phosphorylation the effects could be different. 

-Have you checked the effects of DPHC on inflammation?

-Have you checked the effects of DPHC on insulin sensitivity? Sometimes when adipogenesis is affected, there’s also an effect on insulin sensitivity. Analysing insulin action through the measurement of the ratio pSer473 of Akt/Akt would be highly recommended to pave the way for future studies in mice taking into account the effects on insulin action.

-I would highly recommend enlarging discussion. The manuscript is highly interesting but it would be appreciate more discussion about future research based on the usage of this drug.

Author Response

Reviewer #2: I really appreciate the huge effort of the authors for the composition of this manuscript. It deals with the high potential of the role of DPHC on adipogenesis. However, to strengthen these findings, there are a few questions/suggestions that have arisen throughout the reading.
Response: Thank you very much for spending your valuable time in assessing our manuscript. We appreciate you detailed review and salient comments. We have carried out necessary modifications to the manuscript based on your comments.

Comment 1: The authors describe in Figure 1 the number of replicates and the data representation of the graphs. It would be proper to do the same with other Figures. How do you calculate SE? Do you mean SEM or SD?
Response: We appreciate your suggestion and comment. This study used the standard deviation (SD) equation for data calculate.

Comment 2: In Figure 2, you have analysed Resistin, but you don’t comment this result anywhere.
Response: We have added that mentions in results in page 3 line 85-87.

Comment 3:  Figure 3, is it said that DPHC suppresses the activation of p-AMPKa and p-ACC, when it should be “activates”. Moreover, you should clarify that pAMPK phoshorylates ACC inactivating it, and thus blocking downstream effects. This statement is not clear neither in the results nor in discussion
Response: Thank you for your observation. According to it we have added the discussion in page 4 line 133-140.

Comment 4: Readers should have references for all antibodies used, and phosphorylations analysed, since depending on the phosphorylation the effects could be different.
Response: We have added that references for all antibodies in materials and methods in page 5 line 153-159.

Comment 5: Have you checked the effects of DPHC on inflammation?
Response: We couldn’t perform the experiments on anti-inflammation effect in this manuscript. However, Han et al reported that anti-inflammatory effect of diphlorethohydroxycarmalol (DPHC) isolated from lshige okamuarae in vitro and in vivo.
Han, S.C,; Kang, N.J,; Kang, G.J,; Koh, Y.S,; Hyun, J.W,; Lee, N.H,; Kang, H.K,; Yoo, E.S. Anti-inflammatory effect of diphlorethohydroxycarmalol (DPHC) isolated from Ishige okamuarae in vitro and in vivo. Cytokine. 2014, 70(1), 44-45.  

Comment 6: Have you checked the effects of DPHC on insulin sensitivity? Sometimes when adipogenesis is affected, there’s also an effect on insulin sensitivity. Analysing insulin action through the measurement of the ratio pSer473 of Akt/Akt would be highly recommended to pave the way for future studies in mice taking into account the effects on insulin action. I would highly recommend enlarging discussion. The manuscript is highly interesting but it would be appreciate more discussion about future research based on the usage of this drug.
Response: Many Thanks for your valuable comments and suggestions which have led to significant improvement on the presentation and quality of this paper. Further, we included more discussion on the role of AMPK in adipogenesis by referring to the paper that you have recommended. In addition, we included some more details to the introduction about adipogenesis and involvement of some adipogenic-specific factors. We sincerely hope that our findings will help to enrich the natural product knowledge base and will increase the interest in exploring these compounds in detail. We have strongly considered your suggestions and have provide answers to each question. After further study, we will investigate the anti-obesity effects and insulin sensitive effects of isolation DPHC compounds in high fat diet mice and type 2 diabetic mice.

Thank you in advance for your cooperation. I look forward to receiving your kind response.
Sincerely
Seung-Hong Lee